# Heterologous DNA–Adenovirus Prime–Boost Strategy Expressing Bluetongue Virus VP2 and VP7 Proteins Protects Against Virulent Challenge

**DOI:** 10.3390/vaccines13090991

**Published:** 2025-09-22

**Authors:** Pablo Nogales-Altozano, Laro Gómez-Marcos, Ana Belén Carlón, Andrés Louloudes-Lázaro, Alicia Rivera-Rodríguez, Jaime Larraga, Pedro J. Alcolea, Ana Alonso, Vicente Larraga, Verónica Martín, José M. Rojas, Noemí Sevilla

**Affiliations:** 1Centro de Investigación en Sanidad Animal, Instituto de Investigación y Tecnología Agraria y Alimentaria, Consejo Superior de Investigaciones Científicas (CISA-INIA-CSIC), Valdeolmos, 28130 Madrid, Spain; pablo.nogales@inia.csic.es (P.N.-A.); laro.gomez@inia.csic.es (L.G.-M.); anabelen.carlon@inia.csic.es (A.B.C.); andres.louloudes@inia.csic.es (A.L.-L.); alicia.rivera@inia.csic.es (A.R.-R.); veronica.martin@inia.csic.es (V.M.); 2Centro de Investigaciones Biológicas Margarita Salas (CIBMS-CSIC), 28040 Madrid, Spain; jlarraga@cib.csic.es (J.L.); pjalcolea@cib.csic.es (P.J.A.); amalonso@cib.csic.es (A.A.); vlarraga@cib.csic.es (V.L.)

**Keywords:** orbivirus, T cell response, neutralization, DNA vaccine, adenovirus vaccine, protection

## Abstract

Background/Objectives: Bluetongue virus (BTV) is an emerging arbovirus causing significant economic losses in the ruminant industry. Current vaccines offer limited cross-protection against heterologous serotypes and do not enable differentiation between infected and vaccinated animals (DIVA). Subunit-based vaccines provide a potential DIVA-compatible solution. This study aimed to develop a vaccination protocol expressing BTV structural proteins VP7 or VP2 using antibiotic-resistance-free DNA plasmids and replication-defective adenovirus vectors. Methods: We evaluated homologous DNA prime–boost and heterologous DNA prime–adenovirus boost strategies in a murine model, assessing adaptive immune responses and protection against virulent BTV challenge. Results: The heterologous DNA–adenovirus prime–boost strategy expressing both antigens conferred full protection, preventing viremia, while homologous DNA-DNA prime–boost provided only partial protection. Both VP7 and VP2 elicited cellular and humoral immune responses, but the heterologous strategy significantly enhanced anti-BTV IgG, neutralizing antibody titers, and T cell activation. CD8^+^ T cell responses showed the strongest correlation with viral load reduction, suggesting that cellular immunity to conserved VP7 could serve as a platform for cross-protection against multiple BTV serotypes. Conclusions: These findings highlight the potential of heterologous DNA–adenovirus vaccination as an effective DIVA-compatible strategy for BTV control. By inducing strong and protective immune responses, this approach could improve disease surveillance and management, ultimately reducing the impact of BTV on livestock industries.

## 1. Introduction

Bluetongue virus (BTV) produces a debilitating ruminant disease that must be notified to the world organization for animal health (WOAH) and that leads to significant economic losses through loss of productivity. The disease is characterized by fever, apathy, loss of appetite, and edemas of the face and lips. In the most severe cases, it generates respiratory distress and hemorrhages that can result in the characteristic cyanotic tongue that gives its name to the disease, and even lead to the death of animals [1]. It can also produce abortions in pregnant ewes, which further contribute to the loss of productivity [2]. Outbreaks in disease-free regions can lead to important economic losses [3,4], and it is estimated that USD 3 billion is lost in the industry yearly as a result of BTV infections [5]. Typically sheep are more prone to develop severe clinical signs than other ruminants, but some serotypes can also severely affect cattle, such as in the case of the 2006 BTV-8 outbreak in Europe [6].

BTV is a prototypical member of the genus *Orbivirus*, family *Sedoreoviridae*, that includes other relevant pathogens in animal health such as African horse sickness virus (AHSV) or epizootic hemorrhagic disease virus (EHDV) [7]. It possesses a genome that consists of 10 segments that encodes for 7 structural proteins (named VP1 to VP7) and 4-5 non-structural proteins (named NS1 to NS5) [8,9]. BTV is principally transmitted through the bites of infected *Culicoides* spp. to the mammalian host [2,10]. Viremia is characteristically long in BTV infection, which probably facilitates the transmission of the virus back to the *Culicoides* vector [11,12]. Global warming, which has increased the habitat range of traditional competent vectors, as well as the likely adaptation of the virus to the *Culicoides* species present in more temperate regions, has led to the establishment of BTV in Europe [13]. To further complicate disease control, BTV also possesses multiple serotypes (at least 29 have been described to date) that offer little heterologous cross-serotype protection. Although current inactivated vaccines are effective for homologous serotype protection, they only offer little protection against different serotypes. These vaccines cannot differentiate between infected and vaccinated animals, the so-called DIVA approach. DIVA vaccines are highly desirable in veterinary medicine, as they would allow for differentiating vaccinated animals from infected ones, thus helping in disease control and allowing animal trade to resume in disease-affected areas. Vaccines based on the expression of BTV subunits have the potential to be DIVA vaccines, as they only express parts of the virus. Much effort has been made to develop these subunit vaccines against BTV, either through the expression of recombinant BTV proteins or by expressing the coding sequences of these subunits in recombinant viral vaccine platforms [14]. Some of these approaches have also shown promising protection results as multiserotype vaccines [15,16,17]. These are highly sought after in the field, as production of a single vaccine formulation could be sufficient to protect against BTV in regions in which multiple serotypes circulate.

Protection against BTV infections necessitates a humoral and a cellular response [18,19]. Early depletion and transfer studies demonstrated that solely humoral or cellular responses only offer partial protection against infection [18,19]. Vaccination should therefore aim at eliciting both arms of the adaptive immune response. Cellular immunity is particularly relevant in the case of BTV infections as it can be targeted by subunit vaccines to conserve antigens between serotypes such as VP7 and NS1 [20,21]. Indeed, we have found that expressing VP7 in a recombinant replication-defective adenovirus platform could confer BTV protection in sheep in the absence of neutralizing antibody [22], while cellular responses to an NS1 epitope can also be protective in a murine model [23]. In most cases, cellular immune response stimulation is unlikely to confer sterile protection against BTV [22], and potent humoral antigens such as VP2, which contain the majority of neutralizing determinants, should be included in vaccines.

In the present work, we evaluated the immunogenicity and protective potential of a DNA plasmid platform (pPAL) that lacks a resistance gene for selection and that the European Medicines Agency recently approved for leishmania vaccination in dogs [24]. Expression of the SARS-CoV2 spike and nucleoprotein in this DNA platform was also an effective vaccine in animal models against virulent challenge with SARS-CoV2 [25]. As a proof of principle, we chose to express the highly conserved antigen VP7 and the main determinants for neutralizing antibodies, VP2, from BTV-8 in this DNA expression system. We used a prime–boost regime, either with two inoculations of the pPAL plasmids, or with pPAL plasmid priming followed by a recombinant replication-defective adenovirus booster that encoded the same viral antigens. We evaluated the responses to vaccination and virulent BTV-8 challenge using only VP7 or VP2 as antigens or with a combination of both antigens.

## 2. Material and Methods

### 2.1. Ethical Statement

All the animal experiments were carried out in a disease-secure isolation facility (BSL3) at the Centro de Investigación en Sanidad Animal (CISA), in strict accordance with the recommendations in the guidelines of the Code for Methods and Welfare Considerations in Behavioral Research with Animals (Directive 86/609EC; RD1201/2005), and all efforts were made to minimize suffering. Experiments were approved by the Committee on the Ethics of Consejo Superior de Investigaciones Científicas (CSIC) and the National Animal Welfare Committee (PROEX 295.6/21).

### 2.2. Cell Lines and Viruses

HEK293 (ATCC CRL-1573) cells were grown as described in [26]. BTV-8 (NET2006/04) was grown in BHK-21 cells (ATCC CCL-10) and titered in Vero cells (ATCC CCL-81) by standard plaque assays, as described in [20,27,28]. The construction, preparation and purification of the recombinant replication-defective human adenovirus 5 (Ad) AdVP7, AdVP2 (both antigens from BTV-8), and AdDsRed (control adenovirus that encodes a red fluorescent protein) used in the present work have been described in [22]. BTV-8 was inactivated (iBTV) with binary ethylenimine (BEI) as described in [20,27].

### 2.3. Generation of pPAL-VP7 and VP2 Plasmids

BTV-8 VP7 (KU569990 to KU569999) and VP2 (AB037932.1) sequences were optimized by the Monte Carlo approach according to relative codon usage frequencies [29], synthesized, and cloned into the pPAL expression plasmid that lacks an antibiotic resistance gene as described [25]. Plasmid DNA (pPAL-empty, pPAL-VP7 and pPAL-VP2) was obtained by endotoxin-free giga plasmid (Qiagen, Hilden, Germany).

### 2.4. Immunofluorescence Study of VP7 and VP2 Expression

HEK293 cells were seeded in 12-well plates containing coverslips and subsequently transfected using transIT-LT1 reagent (Mirus Bio, Madison, WI, USA). After 48 h, cells were fixed using 4% paraformaldehyde for 20 min, washed 3 times with PBS and used for immunofluorescence studies. Cells were permeabilized with PBS+ 0.1% (*v*/*v*) triton X-100 (Sigma, Madrid, Spain) for 10 min at room temperature. Preparations were blocked for 45 min at room temperature with Dako background-reducing antibody diluent (Agilent, Santa Clara, CA, USA), and subsequently incubated with primary antibodies overnight at 4 °C (anti-VP7 sheep sera or anti-VP2-8 rabbit sera obtained in-house from adenovirus-vaccinated animals used at a 1:500 dilution [15]). Cells were washed in PBS and stained with highly cross-adsorbed goat anti-sheep/rabbit Alexa Fluor 488 secondary antibody (Thermofisher, Madrid, Spain) and counterstained with 4′,6-diamidino-2-phenylindole DAPI (Sigma, Madrid, Spain) to visualize nucleic acids. Coverslips were mounted with Prolong mounting media (Thermofisher. Madrid, Spain) and confocal images were acquired using an LSM 880 confocal microscope (Zeiss, Madrid, Spain). Image analysis was performed with ImageJ software (v1.54f) (https://imagej.net/ij/ (accessed on 18 September 2025) US National Institutes of Health, Bethesda, MD, USA).

### 2.5. Mouse Vaccination, Serum Preparation and BTV-8 Challenge

This study was approved by the institution ethical committee (PROEX 295.6/21). All steps were taken to minimize animal suffering and animals were assessed daily for clinical signs once infected and euthanized once they reached the defined humane endpoint. To determine vaccine immunogenicity and protection from viral challenge, the IFN-α/βR^0/0^ (IFNAR^(−/−)^) mouse model lacking a type I IFN receptor was chosen given their susceptibility to BTV infection [28]. IFNAR^(−/−)^ mice on a C57BL/6 genetic background were kindly provided by Professor R. Zinkernagel (Institute of Experimental Medicine, Zurich) and bred in our animal facility. For priming immunizations, 8- to 10-week-old female IFNAR^(−/−)^ mice were inoculated with 40 μg of pPAL plasmid intramuscularly followed by an immediate electroporation cycle [25]. Briefly, after plasmid intramuscular inoculation in the hind leg, needle electrodes were placed equidistant to the inoculation point in the direction of the muscle fibers, leaving ~5 mm separation between them, and six 50 ms 100 V square wave pulses were applied in 1 s intervals. The procedure was conducted under gaseous sedation (isofluorane). For combined VP7 and VP2 plasmid inoculations, 40 μg of each pPAL plasmid was administered. After 15 days, mice were boosted with the same antigens, either administered as pPAL plasmids as indicated above or by intramuscular inoculation of a 10^9^ IU recombinant replication-defective human adenovirus 5 (Ad) that encoded the antigen. For the combined VP7 and VP2 adenovirus booster, mice received 10^9^ IU of each adenovirus construct. Mice were randomly separated in 8 groups: 2 control groups that received homologous pPAL-empty (pPAL) prime–boost or heterologous pPAL prime and AdDsRed boost (*n* = 8 per group); 2 groups that received solely the BTV antigen VP7, administered as pPAL-VP7 prime–boost or as pPAL-VP7-prime and AdVP7-boost (*n* = 10 per group); 2 groups that received solely the BTV antigen VP2, administered as pPAL-VP2 prime–boost or as pPAL-VP2-prime and AdVP2-boost (*n* = 10 per group); and 2 groups that received a combination of VP7 and VP2 antigens, administered as a combined pPAL-VP7 and pPAL-VP2 plasmids prime–boost or as combined pPAL-VP7 and pPAL-VP2 plasmids prime and a combined AdVP7 and AdVP2 boost (*n* = 10 per group). Animals were challenged with BTV-8 (10^3^ pfu), given intraperitoneally 15 days after the booster vaccination. Clinical signs were scored as described in Appendix A. To obtain serum for humoral immunity assays, mouse blood was collected by submandibular venipuncture prior to immunization, booster immunization, challenge, and at the end of the experiment in surviving animals. Blood was allowed to clot at room temperature for 30 min in collection tubes and spun down at 10,000× *g* to extract serum. Serum was stored at −80 °C until use. To evaluate viremia, a drop of blood was obtained in EDTA-containing tubes prior to challenge and at days 3, 5, 7 and 12 post-challenge. EDTA–blood was stored at −80 °C until use.

### 2.6. BTV Viremia Detection by RT-qPCR

RNA was obtained from EDTA blood using the IndiSpin pathogen kit (Indical) following the manufacturer’s instructions. RNA was stored at −80 °C until use. BTV viremia was assessed as described using 50 ng of isolated RNA [30]. Data are presented as the *Ct* value of the amplified BTV segment 5 (S5). Detection limit for the assay was *Ct* < 36.

### 2.7. Splenocyte Isolation

Splenocytes were isolated as described in [27]. Briefly, spleens were obtained from animals sacrificed at day 7 post-boost (5 animals per vaccinated groups and 3 per control groups) and disaggregated using a 50 μm cell strainer using complete splenocyte media (RPMI supplemented with 10% FBS (Lonza, Basel, Switzerland), 4 mM L-glutamine, 10 mM HEPES, 1% 100× non-essential amino acids, 1 mM sodium pyruvate, 100U/mL penicillin/100 μg/mL streptomycin and 50 nM β-mercaptoethanol (all from Gibco, Invitrogen, Madrid, Spain)). Erythrocytes were lysed by incubation for 15 min at room temperature with lysis buffer (155 mM NH_4_Cl, 12 mM NaHCO_3_, 140 μM EDTA, pH 7.4). After two further washes in complete splenocyte media, cells were counted and ready for use in ELISpot assays and flow cytometry-based assays.

### 2.8. Mouse IFN-γ ELISpot Assay

Sterile ELISpot plates (MSIPS4510 plate, Millipore, Darmstadt, Germany) were activated with 35% sterile ethanol, and washed once with sterile water and twice with sterile PBS prior to coating with anti-mouse IFN-γ capture antibody (Mouse IFN-γ ELISPOT pair, BD Biosciences, Madrid, Spain). Plates were then blocked in complete splenocyte media for 2 h at room temperature (RT). Freshly isolated splenocytes (2 × 10^5^ per well) were then plated and incubated at 37 °C, 5% CO_2_ for 24 h with BEI-inactivated BTV-8 (iBTV) (equivalent to 10^4^ pfu/well prior to inactivation), a VP7 immunogenic peptide pool (10 μg/mL) (VP7(139) GRWFMRAAQAVTAVV; VP7(324) RPEFAIHGVNPMPGP; VP7(72) AAGINVGPI; VP7(80) ISPDYTQHM; VP7(283) TAILNRTTL; VP7(327) FAIHGVNPM [21]), and concanavalin-A (1.25 μg/mL) as a positive control or they were left unstimulated as negative control. All cultures were performed in triplicates. Cells were then removed and plates washed with PBS + 0.1% Tween (PBS-T) before incubation with biotinylated capture antibody. Plates were washed in PBS-T and incubated with streptavidin-alkaline phosphatase, before being revealed with dissolved SigmaFAST BCIP/NBT tablets (Sigma, Madrid, Spain). Plates were allowed to air dry and spots were counted using an AID ELISpot counter (Autoimmun Diagnostika GmbH, Strassberg, Germany). ELISpot was considered valid when spot counts in unstimulated wells were below 30 and the standard deviation of triplicates was within 25% of the mean. Data are presented as spots above background (i.e., counts in stimulated wells—count in unstimulated wells (mean + 2SD of unstimulated wells)) normalized to 1 × 10^6^ splenocytes.

### 2.9. Flow Cytometry: Intracellular Cytokine Staining (ICS) and T Cell Activation Marker Staining

Freshly isolated splenocytes (1 × 10^6^/well) were stimulated with iBTV or left unstimulated as the control in flat-bottom 96-well plates. Stimulation with an inactivated virus (iBTV) that contains both VP2 and VP7 antigens was chosen for these studies. For intracellular cytokine staining (ICS), cells were incubated for 2 h with iBTV prior to addition of Brefeldin-A (10 μg/mL), Monensin (4 μM), and anti-CD107a antibody (all from Biolegend, San Diego, CA, USA) for a further 4 h. Cells were then stained with viability marker LIVE/DEAD Near IR (Thermofisher, Madrid, Spain) and for surface antigens CD3, CD4 and CD8. Cells were then fixed and permeabilized with a BD Cytofix/Cytoperm kit (BD Biosciences, Madrid, Spain) according to the manufacturer’s protocol, and stained for intracellular cytokines IFN-γ, TNF-α, and IL-2. For T cell activation markers, cells were incubated with iBTV for 8 h in the presence of Monensin and anti-CD154 antibody (both from Biolegend, San Diego, USA). Cells were then placed on ice stained for viability with LIVE/DEAD Near IR marker and for surface markers CD3, CD4, and CD8, and activation markers CD44, CD69 and CD137 (all from Biolegend, San Diego, CA, USA). Cells were then fixed in 4% paraformaldehyde at room temperature for 15 min and washed twice in staining buffer. Samples were acquired on a FACSCelesta flow cytometer (Becton-Dickinson, Madrid, Spain). The gating strategy for ICS has been described in [25]. Fluorescence minus-one (FMO) stainings were included to confirm target cell gating. Analysis was performed with FlowJo software (v10).

### 2.10. Anti-BTV Total IgG, IgG1 and IgG2a/c ELISA

ELISA maxisorp plates (Corning) were coated with iBTV (equivalent to 10^4^ pfu/well) overnight at 4 °C. Plates were then blocked with PBS+5% skimmed milk, washed extensively with PBS + 0.1% tween (PBS-T), and incubated with heat-inactivated sera at different dilutions in PBS + 2% milk for 2 h at room temperature. As a control to evaluate anti-BTV-IgG binding, sera from naïve mice were used. Plates were then extensively washed with PBS-T and incubated for 1 h 30 min with horseradish peroxidase-conjugated anti-mouse IgG antibody (Biorad, Hercules, CA, USA). For IgG1 and IgG2a measurements, plates were coated similarly and blocked with PBS + 5% BSA. Sera were diluted in PBS + 2% BSA, and the presence of BTV reactive IgG1 and IgG2a/c was assessed with biotinylated anti-mouse IgG1 or IgG2a/c (both from Biolegend, San Diego, CA, USA), diluted 1:2000 in PBS + 2%BSA, followed by incubation with streptavidin-HRP (Sigma, Madrid, Spain). Plates were washed extensively with PBS-T and ELISA and revealed with TMB substrate (Thermofisher, Madrid, Spain). Reactions were stopped with addition of sulfuric acid (1M) and absorbance was read at 450nm using a FluoSTAR Omega ELISA plate reader (BMG Labtech, Ortenberg, Germany). Anti-IgG titers were calculated as the inverse dilution required to obtain twice the background absorbance for each mouse (i.e., absorbance of 1:200 dilution of naïve sera) [15]. The negative dilution values were given a value of 1 for graphical representation.

### 2.11. BTV Seroneutralization Assays

Heat-inactivated mouse serum dilutions were incubated with 100 pfu BTV-8 for 1 h at 37 °C in 96 flat-bottom well plates, and subsequently, Vero cells (2 × 10^4^ per cells) were plated and incubated for 96 h. Cultures were then fixed with 2% formaldehyde and stained with crystal violet to evaluate cytopathic effects (CPEs). Neutralizing antibody titers were estimated as the serum concentration at which CPEs are in <50% of wells.

### 2.12. Statistical Analysis

All statistical analysis was performed using GraphPad Prism 8 software. Statistical tests used for experimental data analysis are indicated in the figure legends. GraphPad Prism software was used to calculate the Pearson’s correlation coefficient r and their *p*-value by matching BTV S5 *Ct* values and immune parameter values.

## 3. Results

### 3.1. pPAL-Mediated Expression of BTV Proteins VP7 and VP2 in Transfected Cells

The pPAL plasmid platform has been successfully used as a DNA vaccine to express antigens from SARS-CoV2 [25] and leishmania [24], and is able to induce protection after challenge. This platform has the added advantage of not containing an antibiotic-resistance gene for selection, thus making it suitable for approval as a vaccine by regulating authorities. We therefore express the BTV-8 antigens VP7 and VP2 that elicit cellular and humoral response in pPAL to assess the potency of the platform as a BTV vaccine. To determine expression of VP7 and VP2 genes, HEK293 cells were transfected with pPAL-VP7, pPAL-VP2 or pPAL-empty as a control. After 48 h, cells were fixed, and VP7 and VP2 protein expression was assessed by immunofluorescence studies (Figure 1). Both VP7 and VP2 expression was detected with specific antibodies, whereas in control transfected cells with pPAL-empty, no fluorescence was observed, which demonstrates that BTV-VP7 and -VP2 antigens are expressed in transfected cells. pPAL-VP7 and pPAL-VP2 could therefore be suitable antigen delivery platforms.

### 3.2. Heterologous pPAL Prime + Adenovirus Booster Vaccination Protects Against BTV Lethal Challenge

We next designed several vaccination regimes to determine the immunogenicity of VP7 and VP2 expressed using the pPAL platform, either as a homologous pPAL prime–boost vaccine or as a heterologous prime–boost vaccine consisting of a pPAL prime followed by a boost with replication-defective adenovirus that expresses the same antigen. We have previously characterized the adenovirus constructs used in the present work and demonstrated their immunogenicity and capacity to protect against virulent BTV challenge [15,22]. We therefore primed IFNAR^(−/−)^ mice by intramuscular inoculation, followed by immediate electroporation with pPAL constructs expressing VP7, VP2 (derived from BTV-8), a combination of both plasmids (VP7+VP2), or an empty pPAL construct as a control. For booster immunizations, mice either were inoculated with the same pPAL constructs as described above or received an intramuscular injection of a human replication-defective adenovirus 5 (Ad5) construct that expressed the same antigen(s) given for priming. Vaccinated mice were challenged 15 days after booster vaccination with 1000 pfu of BTV-8 and their disease status was monitored daily thereafter. The homologous DNA+DNA vaccination with BTV antigens partially protected mice against BTV challenge (Figure 2A). The presence of VP2 in the DNA vaccine regime improved survival when compared to only VP7. Mice that received the heterologous vaccination regime consisting of pPAL prime followed by Ad5 booster expressing BTV antigens were fully protected. All control mice that received immunizations that did not include BTV antigens (i.e., pPAL+pPAL or pPAL+Ad5DsRed) succumbed to the disease. We also assessed viremia at several timepoints post-challenge by RT-qPCR (Figure 2B–D). In accordance with the survival data, viremia was controlled by vaccination in heterologous pPAL+Ad5 groups that express BTV antigens, which resulted in the survival of all mice. In the heterologous pPAL+Ad5 that received both BTV antigens, we did not detect viremia, indicating that protection could be sterile in this group. Homologous pPAL vaccination only controlled replication in some individuals, which ultimately led to their survival. Viremia data shows that BTV replicated quickly in control groups, which resulted in the death of the animals. The delivery of VP2 and VP7 through a heterologous DNA + adenovirus regime is therefore sufficient to drastically reduce viral replication and protect IFNAR^(−/−)^ mice from lethal BTV challenge.

### 3.3. Heterologous pPAL Prime + Adenovirus Booster Vaccination Induces Potent Humoral Responses

To identify the correlates of protection with adaptive immunity elicited by vaccination, we first assessed humoral responses triggered by the different vaccine regimes. We measured the presence of anti-BTV IgG in the serum of immunized mice after booster vaccination (Figure 3A). All vaccination regimes, which included VP7 and/or VP2, led to the generation of anti-BTV IgG when compared to controls that did not receive BTV antigens. Booster immunization with adenovirus improved antibody titers for the groups vaccinated with VP7 alone and the group that received VP7+VP2 as antigens. No significant differences in antibody titers were detected in the VP2-alone-vaccinated groups. We next evaluated whether vaccination preferentially elicited IgG1 or IgG2a/c BTV-specific antibodies. This immunoglobulin class switch in mice is indicative of the Th1/Th2 bias in the immune response [31]. Overall, we found that after an adenovirus booster, most mice possessed IgG2a/c rather than IgG1 BTV-specific antibodies, which indicates that the heterologous DNA–adenovirus prime–boost is likely to promote a Th1 bias of the immune response. We also determined the presence of neutralizing antibodies (NAbs) to BTV, since their induction is important for disease control [19]. As predicted, neutralizing antibodies were detected in groups immunized with the VP2 antigen (VP2 alone and VP2+VP7) since the majority of neutralizing determinants are directed to this antigen (Figure 3B). Adenovirus booster vaccination nonetheless significantly improved anti-BTV NAb titers when compared to homologous pPAL vaccination. Overall, these data indicate that adenovirus booster vaccination improves the quality of the antibody response to BTV, as shown by the increase in anti-BTV NAb titers.

### 3.4. Heterologous pPAL Prime + Adenovirus Booster Vaccination Induces Potent Anti-BTV Cellular Responses

We employed several strategies to study the cellular responses elicited by vaccination. We first analyzed T cell activation in response to iBTV stimulation by assessing the expression of the antigen-specific activation markers CD154 on activated CD44^+^ CD4^+^ T cells and CD137 on CD44^high^ CD8^+^ T cells by flow cytometry (Figure 4A). These markers were chosen as they are transiently expressed on T cells in response to antigen stimulation [32,33]. Further analysis of CD137^+^ CD44^high^ CD8^+^ T cells with antigen activation marker CD154 and activation marker CD69 confirmed that this CD8^+^ T cell population represented antigen-activated CD8^+^ T cells (Appendix A). Upon iBTV in vitro stimulation, we detected a significant increase in CD154^+^ cells in activated CD4^+^T cells for all vaccination regimes, which indicates that immunization elicited a BTV-specific CD4^+^ T cell response (Figure 4B). We did not detect a significant increase in CD137^+^ cells in activated CD8^+^ T cells in the pPAL-VP7 + pPAL-VP7 or in the pPAL-VP7+VP2 + pPAL-VP7+VP2 groups (Figure 4C). The presence of these activated CD8^+^ T cells significantly increased for the other vaccination regimes stimulated with iBTV. Importantly, the presence of CD154^+^ CD44^+^ CD4^+^ T cells and CD137^+^ CD44^high^ CD8^+^ T cells was significantly higher in heterologous prime boost groups when compared to homologous DNA groups. Overall, our data indicate that heterologous immunization with pPAL and adenovirus increased the number of BTV-specific T cells when compared to homologous pPAL vaccination.

We also analyzed the cellular responses elicited by the different vaccination regimes using IFN-γ ELISpot assays by measuring the anti-BTV cellular response using iBTV or a pool of immunogenic VP7 peptides [21] in splenocytes 7 days after booster immunization (Figure 5A,B). All vaccination regimes that included BTV antigens induced cellular responses to iBTV, although the responses in the two groups that only received VP7 did not reach significance. Booster vaccination with AdVP2 significantly improved cellular responses to iBTV. When responses were assessed with the VP7 peptide pool, we could only detect potent induction of anti-VP7 cellular immunity in the group that used AdVP7 as booster immunization. Adenovirus-based booster vaccination thus significantly improved cellular responses to VP7 when compared to the homologous pPAL strategy. We next used intracellular cytokine staining (ICS) for IFN-γ, TNF-α, and IL-2 and staining of CD107a as a surrogate marker for cytotoxicity and flow cytometry analysis to determine which T cell populations were activated by immunization (Figure 5C,D). We evaluated in CD4^+^ and CD8^+^ T cells the expression of at least one functional marker (IFN-γ, TNF-α, IL-2, or CD107a) in response to iBTV stimulation. We found that all vaccination regimes induced CD4^+^ and CD8^+^ T cells that could specifically recognize iBTV. Adenovirus booster immunization significantly increased CD4^+^ and CD8^+^ T cell responses to iBTV when compared to homologous pPAL immunization, independently of the given antigen. In line with the T cell activation data (Figure 4), we found in ELISpot and ICS studies that heterologous immunization with pPAL and adenovirus increased the number of T cells responding to BTV stimulation when compared to homologous pPAL vaccination.

Finally, we analyzed the polyfunctionality of anti-BTV CD4^+^ and CD8^+^ T cells by analyzing the concomitant production of IFN-γ, TNF-α, and IL-2, and the expression of the degranulation marker CD107a in flow cytometry (Figure 6). In CD4^+^ T cells, we found that homologous pPAL immunization could elicit potentially cytotoxic CD4^+^ T cells (CD107a^+^) (Figure 6A). We also detected CD4^+^ T cells expressing TNF-α in most animals but low levels of IL-2 and IFN-γ production with homologous pPAL BTV vaccination. Heterologous immunization increased the percentage of CD4^+^ T cells producing these cytokines to iBTV, but did not elicit significant levels of cytotoxic CD4^+^ T cells, as measured by CD107a expression. Importantly, we detected IFN-γ-producing CD4^+^ T cells for iBTV with heterologous vaccination regimes. In CD8^+^ T cells, homologous pPAL immunization only elicited low levels of IFN-γ-, TNF-α-, and IL-2-producing cells (except for two mice in the VP2-immunized group) (Figure 6B). Heterologous immunization with pPAL and adenovirus increased the percentage of BTV-specific cytotoxic CD8^+^ T cells, as well as that of cytokine-producing cells responding to iBTV when compared to homologous pPAL vaccination. We also evaluated the capacity of CD4^+^ and CD8^+^ T cells to simultaneously express multiple cytokine/degranulation markers as a measure of the T cell polyfunctionality elicited by the vaccine (Figure 6C,D). Most vaccination regimes elicited limited polyfunctional T CD4^+^ or CD8^+^ T cells, but inclusion of VP7 delivery through the adenovirus booster (pPAL-VP7 + AdVP7 and pPAL-VP2+VP7 + AdVP7+VP2 groups) increased the percentage of polyfunctional T cells when compared to the homologous DNA boost. Overall, our study indicates that heterologous pPAL + adenovirus vaccination with BTV antigens VP7 and VP2 improves T cells responses to the virus, not only quantitatively, but also qualitatively.

### 3.5. Vaccine-Induced Adaptive Immune Response Parameters Correlate with Protection

We wanted to determine which parameters of the adaptive immune responses elicited by vaccination would correlate with protection, as analyzed by others [34]. We therefore calculated the Pearson’s correlation coefficients *r* between adaptive immune measurements and viremia at days 3 and 5 pi (Table 1). We found a significant positive correlation (*r* > 0.5; *p* < 10^−3^) between all immune parameters and *Ct* values for BTV-S5; i.e., immune parameters correlated with a reduction in viral load. This confirms that induction of adaptive immune responses to BTV by the vaccine is associated with a reduction in viremia at days 3 and 5 pi that ultimately leads to recovery from the disease. It is worth noting that activation of CD8^+^ T cells (as measured by cytokine secretion and degranulation) displayed the highest correlation with reduced disease burden. The presence of neutralizing antibodies to BTV was also a higher correlate of reduced disease than anti-BTV IgG. Overall, these data indicate that BTV vaccines inducing cellular and humoral immunity are likely to protect adequately against BTV.

## 4. Discussion

In the present work, we show that delivery of BTV antigens VP7 and VP2, through DNA priming followed by an adenovirus booster, can fully protect mice against virulent BTV challenge. Heterologous prime–boost strategies have been used in the past and shown promising results in protecting against BTV [16,17,35,36,37]. BTV antigen delivery through DNA prime followed by a modified vaccinia virus Ankara (MVA) booster can protect mice from BTV lethal challenge [16,35]. Similarly, antigen delivery through avian reovirus microspheres followed by an MVA booster was also successful for protection in murine models [36]. Heterologous strategies involving two viral vectors (ChAdOx1 vector and MVA) have also shown some efficacy in mice and sheep [17,37]. Our strategy nonetheless makes use of two delivery systems (pPAL and adenovirus platforms) that have been approved as vaccines by regulatory agencies. We also present an exhaustive assessment of the immune response generated by these heterologous strategies as well as their correlation with protection. We found that in the murine model, induction of anti-BTV CD8^+^ T cells showed the highest correlation coefficient with protection. We also found that neutralizing antibody induction correlated strongly with protection. Inducing responses that include both arms of adaptive immunity would indeed mimic immune responses to BTV in the natural host [18,19]. Vaccination should therefore aim at stimulating both immune parameters, and vaccine potency is likely to rely on both parameters.

In support of this, we found that viremia was better controlled in groups that received VP2 than in the group that only received VP7. This is likely due to the presence of anti-VP2 neutralizing antibodies to BTV in these animals that quickly limit replication, whereas anti-VP7 antibodies cannot effectively neutralize BTV infection. We nonetheless detected low NAb titers in one mouse immunized solely with VP7 following adenovirus boost. Although antibodies to the BTV-VP7 inner core protein are not thought to produce neutralization, reports exist of neutralizing determinants on the VP6 inner core proteins of the rotavirus [38,39], another member of the *Sedoreoviridae* family. Inner core proteins in *Sedoreoviridae* are mainly exposed intracellularly during infection [8]; thus, antibodies are unlikely to “classically” neutralized infection by preventing virus entry. It is more probable that neutralization against VP7 determinants would occur through intracellular mechanisms mediated, for instance, by antibody binding to the intracellular IgG receptor TRIM21 [40]. The observation that some neutralizing antibodies can be elicited against BTV-VP7 suggests that a neutralizing determinant could also exist on this conserved BTV protein. Further work will be required to elucidate this observation.

Another important aspect that is critical for BTV protection is the choice of antigen to include in the vaccine formulation. Our rationale was to focus on structural proteins as antigens, since they are present on the viral particles, and this should allow for quicker recognition and elimination of the virus and infected cells when compared to non-structural proteins, which require viral replication to occur for expression to begin. We chose VP7 and VP2 as antigens, as they are abundant proteins on the virion (260 and 60 copies, respectively [8]) and therefore their availability for presentation to the immune system as a viral antigen should be high during infection. Moreover, they are also known to contain cellular and humoral determinants [21,41]. We have found that VP7 could represent the basis for cross-serotype protection and that this is likely due to cellular immunity, since we detected protection in mice and sheep in the absence of neutralizing antibodies [15,22]. Other studies also confirmed that induction of immunity to VP7 participates in protection [35,36]. Indeed, the conservation of VP7 T cell epitopes across different BTV serotypes could allow for cross-protection [21]. The non-structural proteins NS1 and NS2 have also been used as immunogens in other studies to trigger cellular immunity. These non-structural proteins are conserved between serotypes and can therefore be used to elicit serotype cross-reactive T cells [17,20,23]. Vaccination with these antigens can lead to protection in animal models, but protection was limited in sheep, possibly because only targeting these antigens is insufficient to control viral replication in the early stages of infection [17,37]. Alternatively, antigen expression and dose optimization by these vaccine platforms could also account for the limited protection afforded in these studies. As previously discussed, VP2 improved protection against virulent BTV challenge in our experiment, and this is likely due to the induction of antibodies that can rapidly neutralize BTV infection. A recent study has corroborated this, showing that inclusion of VP2 in a recombinant MVA with NS1 and a truncated NS2 could offer protection in sheep against BTV challenge, whereas concomitant expression of VP7 with NS1 and the truncated NS2 only offered partial protection [42]. This further indicates that the inclusion of cellular and humoral immune targets of BTV is a requisite to design effective BTV vaccines.

Curiously, we observed that homologous pPAL immunization could potentially elicit anti-BTV cytotoxic CD107a^+^ CD4^+^ T cells. We have previously observed a similar phenotype using pPAL vaccination against SARS-CoV2 [25]. DNA vaccines are known to potentiate cellular responses [43], although this phenotype has not been described before to the best of our knowledge. Whether this observation is specific to the pPAL expression system or can be extended to other DNA vaccine platforms remains to be determined. Recently, using the DNA vaccine platform pVAX, targeting of the Ebola virus glycoprotein to the lysosome pathway through fusion with LAMP-1 led to the activation of CD4^+^T cells [44], which indicates that DNA vaccines can indeed lead to potent CD4^+^ T cell activation. The exact contribution of these cytotoxic CD4^+^ T cells to infection resolution is not fully elucidated [45]. Some studies indicate that they participate in virus clearance [46,47]; however, their presence could also lead to the loss of the B cell germinal center observed in patients that succumbed to SARS-CoV2 infection [48]. Our data nonetheless indicate that BTV is better controlled in this murine model by CD8^+^ T cell activation, and as such, vaccination should aim at triggering this cell population.

Although our data highlights the importance of cellular immunity for protection against BTV in IFNAR^(−/−)^ mice, the immune mechanisms that lead to protection could be different in the natural host. Indeed, a recent study has indicated that CD8^+^ T cells could be detrimental to the bluetongue disease outcome in sheep, possibly through immune-mediated pathogenicity [49]. We have however found in several experimental infections with BTV in sheep that CD8^+^ T cells expand at day 10–15 pi [30,50,51], and this typically coincides with reduced viremia, which points at a role for these T cells in the removal of BTV-infected cells. We nonetheless detected that during the peak of viral replication, T cell responses are acutely suppressed, and this coincided with the overexpression of genes and molecules involved in the immunoregulatory PD1/PD-L1 pathway [30]. In this context, and given the putative role of CD8^+^ T cells in bluetongue immunopathology, it could be hypothesized that in the natural host, these T cells would become activated during acute viral infection to clear infected cells, but their prolonged activation could lead to immunopathology, hence the triggering of immune checkpoints to tightly regulate the activity of the population.

It is also important to note that, although IFNAR^(−/−)^ mice are excellent models to evaluate vaccine candidates against BTV, they are partially immunocompromised. As such, this model has some limitations when evaluating immune responses to vaccines [52,53]. We found that IFN-γ production was quite limited in these experiments when compared to previous work with immunocompetent transgenic mice susceptible to SARS-CoV2 [25]. This could be due to the IFN deficiency of this murine model that could result in impaired IFN type II responses. In comparison, TNF-α responses appear enhanced when contrasted to previous studies in fully immunocompetent mice, which suggest that IFNAR^(−/−)^ mice may compensate for their IFN-I deficiency by promoting pro-inflammatory cytokine production such as TNF-α production. This is an important consideration when using this model to evaluate immune responses to vaccine candidates, given the importance of IFN-I in modulating adaptive immunity [54]. As a result, IFN-I deficiency in these animals could limit and bias the adaptive immunity we observe. In spite of this, we found that IFNAR^(-/-)^ mice mounted T cell responses to a similar repertoire of VP7 epitopes than wild-type mice with the same genetic background [21], and many vaccination studies attest that the immune responses in these mice are sufficient to control viral replication. It would be nonetheless interesting to perform a parallel vaccination study with immunocompetent mice to further evaluate the capacity of the IFNAR^(−/−)^ murine model to mount fully effective adaptive immune responses in future work. These studies could help in the interpretation of immune results in IFNAR^(−/−)^ mice. In spite of these limitations, it is worth noting that IFNAR^(−/−)^ mice are a model with translational potential to the natural host. For instance, several studies have found that VP7 delivery through adenovirus or vaccinia can provide protection in sheep against virulent BTV challenge, thus translating findings from the murine model to the natural host of the disease [22,42].

Finally, our data show that heterologous prime–boost improves the titers of neutralizing antibodies when compared to the homologous DNA vaccination strategy. In contrast, we found in a previous study that a homologous adenovirus expressing VP2 from BTV-1 elicited minimal titers of neutralizing antibodies [15]. This could be due to the difference in VP2 sequence studies or to protein expression levels. Indeed, we have obtained adequate neutralizing antibody titers to other viral antigens, such as the hemagglutinin from peste des petits ruminants virus, with recombinant adenovirus vaccines when using homologous prime–boost vaccination [55], which suggests that this discrepancy may lie in the different VP2 constructs used in these studies. Adequate neutralizing antibody titers against SARS-CoV2 can also be reached with homologous recombinant adenovirus vaccines or with pPAL [25,56]. Indeed, in previous work with homologous adenovirus vaccination against BTV, we found that cross-serotype protection was mostly mediated through cellular immunity [15]. This confirms that adenovirus platforms are therefore potent tools to induce cellular immunity [57]. Nonetheless, one of the main advantages of heterologous prime–boost vaccination over homologous vaccination is that it improves neutralization antibody generation. For instance, in immunocompromised patients, adenovirus prime followed by an mRNA vaccine boost against COVID-19 was shown to improve neutralizing antibody titers [58]. Similarly, we found that a DNA prime–adenovirus boost with BTV antigens improved cellular responses when compared to homologous DNA vaccination. Adenovirus-based VP7 delivery improved the polyfunctionality of anti-BTV T cells, a feature associated with more effective immune responses to pathogens [59]. Evidence exists that heterologous prime–boost strategies also enhance cellular immunity by increasing the number of responding cells and promoting effector memory differentiation [58,60,61,62]. Heterologous prime–boost strategies can therefore be advantageous as they potentiate both arms of adaptive immunity. Although in the context of veterinary medicine, the licensing of two vaccine reagents could be costly, it could still be a useful strategy to explore to produce BTV vaccines that induce long-lasting immunity, particularly if they allow for multiserotype protection.

We herein report a promising strategy for BTV vaccination that consists of a heterologous prime–boost strategy with DNA and adenovirus vectors that express VP7 and VP2 proteins from BTV. This strategy has the potential to reach the clinic given the approval by regulatory agencies of both vaccination platforms in other diseases [24,63]. Expression of both antigens induced potent cellular and humoral immunity that led to protection. Importantly, we could not detect viremia in the vaccinated group that received both antigens, which indicates that this vaccine could produce sterile protection. It will be important in future studies to evaluate the protective efficacy of this heterologous prime–boost regime in a natural host of the disease. Heterologous DNA–adenovirus prime–boost vaccinations are therefore a promising strategy to explore for the development of DIVA BTV vaccines.

## 5. Conclusions

This study demonstrates that a heterologous DNA–adenovirus prime–boost vaccination strategy expressing BTV-VP7 and -VP2 provides full protection against virulent BTV challenge in a murine model. This approach enhances both cellular and humoral immune responses, with CD8^+^ T cell activation and neutralizing antibody titers strongly correlating with viral clearance. VP2 contributed significantly to protection by inducing neutralizing antibodies, while VP7 may also have unrecognized neutralizing determinants. The inclusion of both antigens ensures rapid immune recognition and effective protection. Heterologous prime–boost vaccination proved superior to homologous strategies by enhancing adaptive immunity, making it a promising approach despite potential regulatory challenges. Although IFNAR^(-/-)^ mice are useful for vaccine evaluation, further studies in natural hosts are needed. Overall, this strategy represents a strong candidate for DIVA-compatible BTV vaccination.

## Figures and Tables

**Figure 1 vaccines-13-00991-f001:**
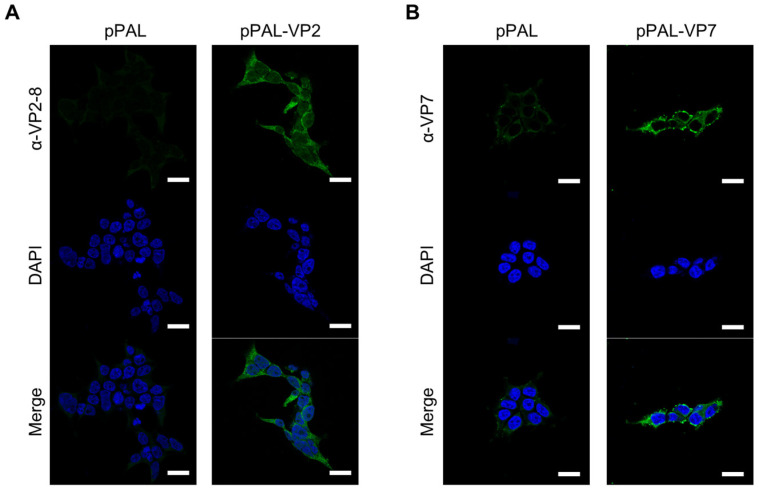
Immunofluorescence studies confirmed VP2 and VP7 expression by pPAL plasmids. HEK-293 cells were transfected with pPAL-VP2 pPAL-VP7, or pPAL-empty as control. After 48 h, cells were fixed permeabilized and stained to detect the expression of VP2-8 or VP7, before counterstaining nucleic acids with DAPI. (**A**) VP2-8 expression in pPAL-empty or pPAL-VP2-8 transfected HEK293 cells. (**B**) VP7 expression in pPAL-empty or pPAL-VP7 transfected HEK293 cells. Scale bar 20 µm.

**Figure 2 vaccines-13-00991-f002:**
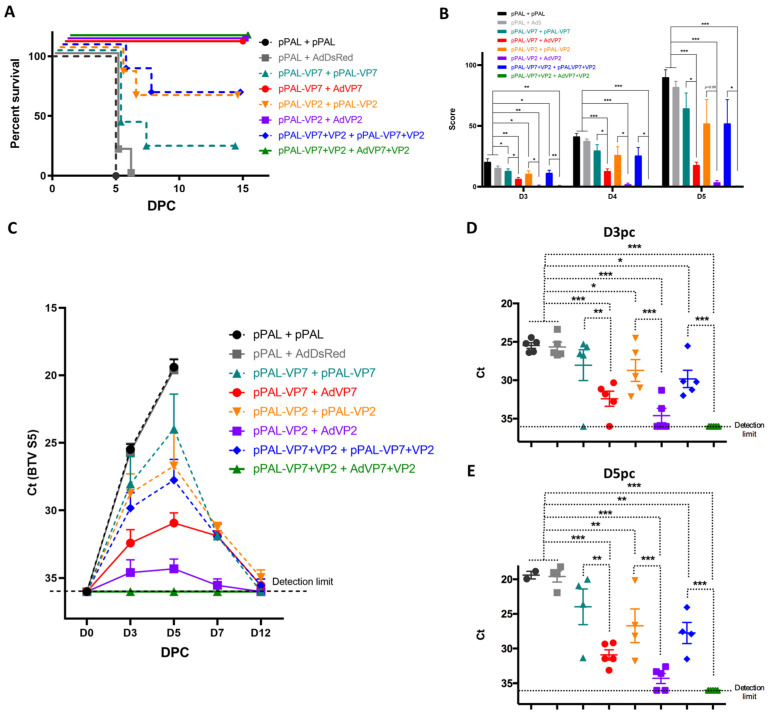
pPAL and adenovirus vaccination protects against BTV-8 challenge and controls viral replication. (**A**) IFNAR^(−/−)^ mice were immunized with VP7 and/or VP2 delivered as homologous pPAL or heterologous pPAL+adenovirus immunizations. Mice (*n* = 5 per group) were challenged with 1000 pfu BTV-8, and monitored daily for appearance of clinical signs of disease. Survival curve for each vaccinated group is shown. (**B**) Clinical scores for each vaccinated group at days 3, 4, and 5 post-challenge are presented. (**C**–**E**) Blood samples were obtained from infected mice at day 0 (pre-challenge) and at days 3, 5, 7 and 12 post-challenge (DPC), and RNA extracted. RT-qPCR was performed to detect BTV RNA presence in these samples. (**C**) *Ct* mean values for BTV segment 5 fragment (BTV S5) are plotted for all timepoints assessed for each treatment group. *Ct* individual values at (**D**) day 3 post-challenge (D3PC) and (**E**) day 5pi (D5PC) are plotted for each vaccinated group. * *p* < 0.05; ** *p* < 0.01; *** *p* < 0.001. One-way ANOVA with Fisher’s LSD post-test. Detection limit of RT-qPCR for BTV S5 fragment (*Ct* < 36) is indicated.

**Figure 3 vaccines-13-00991-f003:**
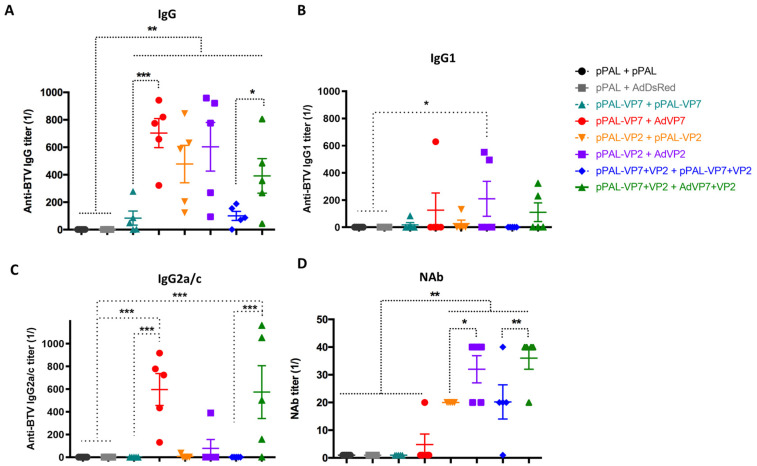
pPAL and adenovirus immunization induces anti-BTV humoral immune responses. Sera from immunized mice were collected 14 days post-booster vaccination and analyzed for the presence of (**A**) anti-BTV IgG, (**B**) antiBTV IgG1, (**C**) anti-BTV IgG2a/c, and (**D**) neutralizing antibodies (NAb) to BTV-8. (**A**–**C**) Anti-BTV IgG/IgG1/IgG2a/c titers (1/) (mean ± SEM) were assessed by ELISA in immunized animals for each vaccination regime. (**D**) NAb titers to BTV-8 (1/) (mean ± SEM) were measured by seroneutralization assays in Vero cells. * *p* < 0.05; ** *p* < 0.01; *** *p* < 0.001. One-way ANOVA with Fisher’s LSD post-test.

**Figure 4 vaccines-13-00991-f004:**
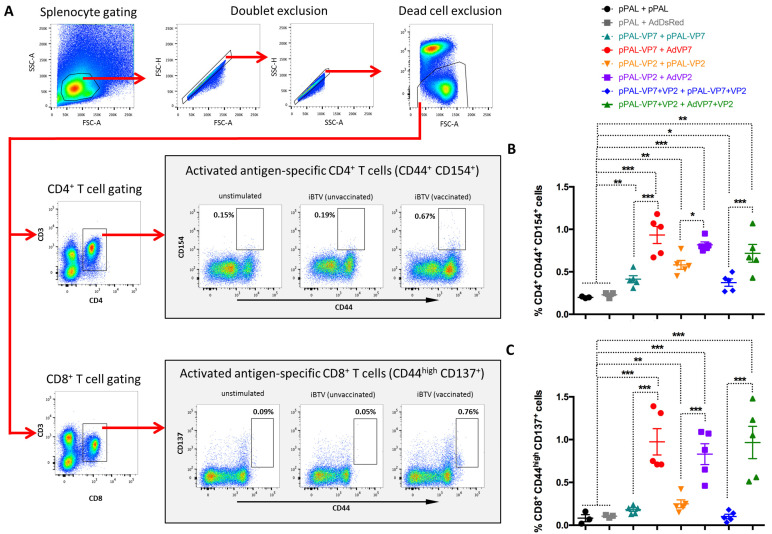
Activation marker induction in CD4^+^ and CD8^+^ T cells stimulated with iBTV. Splenocytes were stimulated with iBTV (or left unstimulated as control) for 8 h and stained for CD3, CD4, CD8, CD44, CD137 and CD154 expression. (**A**) Gating strategy for CD4^+^ CD44^+^ CD154^+^ and CD8^+^ CD44^high^ CD137^+^ cells is shown. CD44^+^ CD154^+^ events for CD4^+^ T cells and CD44^high^ CD137^+^ events for CD8^+^ T cells were considered as activated antigen-specific cells. FSC-A/SSC-A dot-plots were used for splenocyte gating, followed by doublet and dead cell exclusion. CD3/CD4 and CD3/CD8 dot-plots were used for CD4^+^ T and CD8^+^ T cell gating, respectively. Activated antigen-specific cell gating was established using unstimulated splenocytes for each culture as shown. Representatives for iBTV-stimulated are shown in the gating strategy. (**B**) Percentage of CD4^+^ CD44^+^ CD154^+^ cells in iBTV cultures of vaccinated or control mice. (**C**) Percentage of CD8^+^ CD44^high^ CD137^+^ cells in iBTV cultures of vaccinated or control mice. * *p* < 0.05; ** *p* < 0.01; *** *p* < 0.001. One-way ANOVA with Fisher’s LSD post-test.

**Figure 5 vaccines-13-00991-f005:**
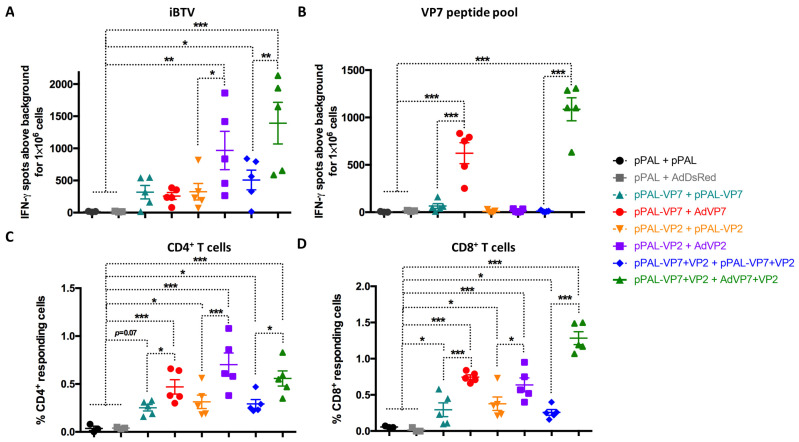
Heterologous prime–boost improves T cell responses to BTV. (**A**,**B**) Splenocytes were seeded in anti-IFN-γ-coated ELISPOT plates and stimulated overnight with (**A**) iBTV or (**B**) a VP7 immunogenic peptide pool. Splenocytes were discarded and ELISPOT plates revealed for IFN-γ secretion. (**C**,**D**) Intracellular cytokine staining (ICS) for CD107a, IFN-γ, IL-2, and TNF-α was performed on iBTV-stimulated splenocytes and marker expression was analyzed by flow cytometry in CD4^+^ and CD8^+^ T cells. Total percentage of (**C**) CD4^+^ or (**D**) CD8^+^ T cells responding to iBTV stimulation (i.e., expressing at least one stimulation marker CD107a, IFN-γ, IL-2, TNF-α) was obtained by Boolean gating in FlowJo using the “OR” function for CD107a-, IFN-γ-, IL-2-, and TNF-α positive events in the CD4^+^ or CD8^+^ gate. * *p* < 0.05; ** *p* < 0.01; *** *p* < 0.001. One-way ANOVA with Fisher’s LSD post-test.

**Figure 6 vaccines-13-00991-f006:**
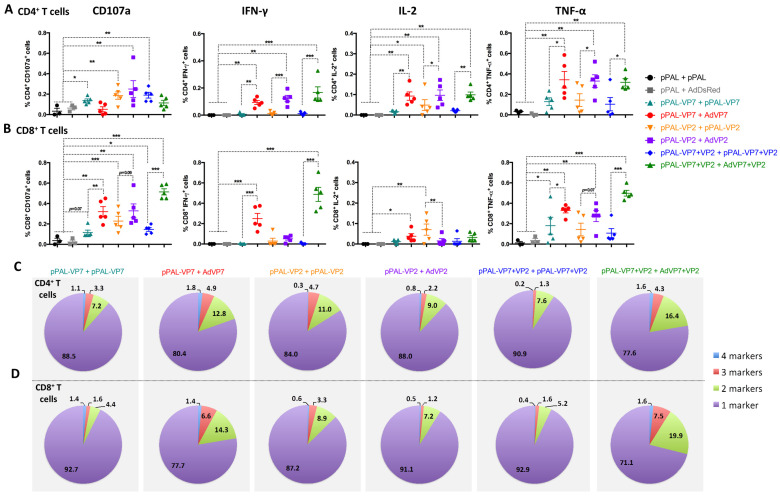
Heterologous prime–boost with VP7 improves the induction of BTV-specific polyfunctional T cells. Splenocytes were stimulated with iBTV, and expression of CD107a, IFN-γ, IL-2, and TNF-α was assessed in CD4^+^ and CD8^+^ T cells by intracellular cytokine staining and flow cytometry analysis. (**A**,**B**) Expression of CD107a, IFN-γ, IL-2, and TNF-α in (**A**) CD4^+^ T cells and (**B**) CD8^+^ T cells. * *p* < 0.05; ** *p* > 0.01; *** *p* < 0.001. One way ANOVA with Fisher‘s LSD post-test. (**C**,**D**) Percentage of simultaneous expression of one, two, three, or four stimulation markers in (**C**) CD4^+^ T cells and in (**D**) CD8^+^ T cells in response to iBTV for each vaccine regimen.

**Table 1 vaccines-13-00991-t001:** Pearson’s correlation coefficients (*r*) between immune parameters induced by vaccination and viral load at days 3 and 5 post-challenge. Pearson’s correlation coefficients *r* were calculated with GraphPad Prism software using *Ct* values for BTV S5 and values of immune parameters. All coefficients had significant *p* values (*p* < 10^−3^).

Immune Parameter	*Ct*(BTV S5) D3pi	*Ct*(BTV S5) D5pi		*p*-Value
IgG titer	0.541	0.611		*p* < 10^−3^
NAb titer	0.671	0.703		*p* < 10^−4^
IFN-γ counts (ELISpot)	0.626	0.682		*p* < 10^−5^
CD4^+^CD44^+^CD154^+^ cells	0.600	0.653		*p* < 10^−6^
CD8^+^CD44^high^CD137^+^ cells	0.696	0.674		*p* < 10^−7^
CD4^+^(CD107a^+^/IFN-γ^+^/TNF-α^+^/IL-2^+^) cells	0.694	0.768		
CD8^+^ (CD107a^+^/IFN-γ^+^/TNF-α^+^/IL-2^+^) cells	0.791	0.814		

## Data Availability

Data available upon request.

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
