# Peer review of "Heterologous DNA–Adenovirus Prime–Boost Strategy Expressing Bluetongue Virus VP2 and VP7 Proteins Protects Against Virulent Challenge"

_vaccines, 2025, doi:10.3390/vaccines13090991_

Round 1

Reviewer 1 Report (Previous Reviewer 2)

Comments and Suggestions for Authors

Authors have successfully addressed immunofluorescence issues regarding VP7 labeling.

Author Response

We would like to thank the reviewer for taking the time to evaluate our manuscript

Reviewer 2 Report (Previous Reviewer 4)

Comments and Suggestions for Authors

I have no other comment

Author Response

We would like to thank the reviewer for taking the time to evaluate our manuscript

Reviewer 3 Report (New Reviewer)

Comments and Suggestions for Authors

The manuscript evaluated the application of pPAL plasmid expressing VP2 and VP7 proteins from bluetongue virus (BTV) serotype 8 as a DNA vaccine. Then immunization schemes in IFNAR(-/-) mice model were performed priming with BTV DNA vaccine in a single approach or by the combination of both antigens. Then, the animal group was boosted with the same antigen in the plasmid vector or adenovirus containing the same antigen or the combination of both antigens in plasmid or adenovirus for the group previously immunized with two antigens. The animal groups were then challenged with virulent BTV-8 and the immunogenicity and the cellular immunity were extensively analyzed.

In summary, the manuscript explored the potential of heterologous DNA-adenovirus vaccination approaches that conferred full protection and prevented viremia, suggesting a promising platform for the cross-protection against multiple BTV serotypes that will be tested in the future study.

In general, the manuscript is well presented and shows promising and novel vaccination strategy. Some comments are listed below for the improvement:

- Original images were provided as .czi files, and it was impossible to open them.

- Lanes 115-116: Add a reference to Monte Carlo approach used for VP2 and VP7 sequence optimization or describe this approach. And how were VP2 and VP7 genes obtained, by synthesis or other methodology?

- Lanes 166-168: Inform how the blood collection was made; would it be via the intravenous route?

- Lanes 173-177: Was the expression level of an endogenous control gene evaluated together with BTV segment 5 gene in RT-qPCR?

- Lane 311: The symbol and color specifications were presented in Fig. 2A and it is unnecessary to show them in other figures.

- Fig. 2B shows clinical scores post-challenge and it is unclear how these parameters were measured because no specification was presented for that.

- Lane 364: Supplementary Fig. 1 cited in the text was unavailable for the review process.

Author Response

Comments 1: Original images were provided as .czi files, and it was impossible to open them

Response: We have now provided the images as .tiff files as cross-platform format.

Comments 2: Lanes 115-116: Add a reference to Monte Carlo approach used for VP2 and VP7 sequence
optimization or describe this approach. And how were VP2 and VP7 genes obtained, by synthesis
or other methodology?

Response: A reference has been added for the sequence optimization for VP2 and VP7 genes. These
sequences were synthesized. This is now stated in M&M.

Comments 3: Lanes 166-168: Inform how the blood collection was made; would it be via the intravenous route?

Response: Blood was collected by submandibular venipuncture, this is now specified in M&M.

Comments 4: Lanes 173-177: Was the expression level of an endogenous control gene evaluated together with BTV segment 5 gene in RT-qPCR?

Response: We did not evaluate the expression of an endogenous gene in these assays. We however used similar volume of blood to extract RNA, and RNA amounts were measured and normalized
to 50ng prior the RT-qPCR. This is now indicated in the manuscript.

Comments 5: Lane 311: The symbol and color specifications were presented in Fig. 2A and it is unnecessary to show them in other figures.

Response: We agree that color and symbol presented on each figure can appear redundant, but we
actually prefer presenting each figure with its legend so that the reader can interpret them
independently.

Comments 6: Fig. 2B shows clinical scores post-challenge and it is unclear how these parameters were
measured because no specification was presented for that.

Response: We have now included a supplementary table S1 with the clinical sign scoring.

Comments 7: Lane 364: Supplementary Fig. 1 cited in the text was unavailable for the review process.

Response: We have now included Supplementary Figure 1 to the manuscript

This manuscript is a resubmission of an earlier submission. The following is a list of the peer review reports and author responses from that submission.

Round 1

Reviewer 1 Report

Comments and Suggestions for Authors

The quality of the manuscript has improved significantly following the revisions.

Reviewer 2 Report

Comments and Suggestions for Authors

In this new draft, authors have successfully addressed most of reviewer comments. However, the most significant issue detected during the first round of revision is still a concern. This is the expression of the viral antigen VP7 by transfection of HEK293 cells with the plasmid encoding VP7 and used for further immunizations. Images provided still do not prove with confidence the expression of VP7 antigen. Based on the literature (just as an example: Monoclonal antibody and B-cell epitope mapping of the VP7 protein in bluetongue virus, by Xin-Bing Hu and others or Expression of VP7, a Bluetongue Virus Group Specific Antigen by Viral Vectors: Analysis of the Induced Immune Responses and Evaluation of Protective Potential in Sheep by Coraline Bouet-Cararo and others), both native VP7 expression from virus infection and recombinant VP7 expression from transfection or viral vector infection show an expression pattern like the one observed with VP2 (which authors clearly demonstrate in this work). VP7 labeling shown here appears to be membrane-associated, with no clear evidence of expression inside of the cell. A commercial monoclonal anti-VP7 antibody (reference: CJ-F-BTV-MAB-10ML, from VMRD) was used for this staining and authors also tried to detect VP7 expression by immunoblot using this reagent, with negative results.

Sometimes, monoclonal antibodies do not work properly under certain conditions. One question that arises now is whether this monoclonal used here has been previously tested for a BTV in vitro infection by IF, to see if it recognizes the native antigen and to determine the optimal concentration of monoclonal used. Trying a battery of different anti VP7 antibodies could help to demonstrate VP7 expression with confidence:

Another alternative is the use of other monoclonal antibodies directed against VP7 (examples are Bluetongue Virus BTV-1 VP7 recombinant monoclonal antibody (20E9) from the European Virus Archive or [9C2C.2] from Creative Diagnostics) or even better, VP7-polyclonal antibodies from serum from BTV-infected or VP7-immunized animals. By using polyclonal abs, both conformational and linear epitopes in VP7 could be detected by both IF and I-blot.

Additionally, and complementary to this IF and I-blot expression analysis, an ELISA test using the serum from the plasmid-VP7 or plasmid-adeno-VP7 immunized mice, coating with recombinant VP7 should be performed, to clearly determine the specific reactivity against this antigen.

Demonstrating the expression of VP7 antigen is crucial to support the conclusions obtained in this immunization work.

As a minor comment, in Figure 3 legend, letters must be re-organized, including D) for the Neutralizing analysis.

Reviewer 3 Report

Comments and Suggestions for Authors

While the authors have addressed some of the concerns, the fact remains that they are reporting outcomes of 1 experiment.  While the authors suggest that this is in line with the ethos of reducing waste and unnecessary animal experimentation, I am not of the opinion that any results should be published that are not replicated at least once.  It is understandable that, especially for clinical studies, sample availability might limit the ability to repeat analysis etc., this is a mouse study using widely available strains, and I am surprised that the authors do not value repeating outcomes, especially as they are trying to substantiate a new vaccine approach.

My biggest concern is ‘rigor and reproducibility’ of the results.  It’s a pretty basic tenant, especially when working with animal models and complex procedures like vaccinations and infections, that the result be repeated at least once, with similar results being obtained. It’s a very dangerous position to take to trust outcomes of a single experiment (and I do not agree with the argument that this line of investigation is guided by adherence to moral obligations to cut animal use – of course we all try to limit animal use, but to do so encompassing the required standards for rigor and reproducibility.

Reviewer 4 Report

Comments and Suggestions for Authors

The manuscript titled *"Heterologous DNA-Adenovirus Prime-Boost Strategy Expressing Bluetongue Virus VP2 and VP7 Proteins Protects Against Virulent Challenge"* presents a well-designed study evaluating a novel vaccination approach against Bluetongue virus (BTV). The work is scientifically sound, methodologically rigorous, and addresses an important gap in veterinary vaccinology by proposing a DIVA-compatible strategy. The data robustly support the conclusions, demonstrating that heterologous DNA-adenovirus prime-boost vaccination induces potent humoral and cellular immunity, correlating with sterile protection in mice. Below are specific comments to further strengthen the manuscript.

1 For Figure 2 Survival Data: Include the number of mice per group in the figure legend (currently only in Methods). Consider adding a panel showing weight loss/clinical scores to complement survival.

2 For Discussion of Cross-Protection (Lines 442–450): Briefly highlight how VP7 conservation across BTV serotypes might translate to broader protection (e.g., cite epitope conservation data from Rojas et al. 2011).

3 For Methods Clarifications: Electroporation parameters (Section 2.4): Specify pulse duration/waveform (e.g., square-wave). ELISpot validation (Section 2.8): Define "background" (e.g., mean + 2SD of unstimulated wells).

4  Line 108: Replace "pyrexia" with "fever" for broader readability.

Line 320: "One-way ANOVA" should specify post-hoc test (e.g., Fisher’s LSD).

5 The finding of cytotoxic CD4 + T cells in pPAL-vaccinated mice is novel but lacks functional validation. Suggest:Include ex vivo cytotoxicity assays (e.g., target cell killing) to confirm their role. Discuss whether this phenotype is unique to pPAL or generalizable to other DNA vaccines (e.g., pVAX).

6 While the IFNAR−/−−/− mouse model is valuable, its immunocompromised nature limits extrapolation to ruminants. The authors acknowledge this but could: Cite preliminary data or prior studies (e.g., sheep adenovirus-VP7 trials from their 2015 paper) to contextualize translational potential.

Propose a timeline/experimental design for future sheep trials, given the regulatory approval of pPAL/adenovirus platforms.